# Protein Ingredient in Saliva on Oral Dryness Patients Caused by Calcium Blocker

**DOI:** 10.3390/geriatrics5040070

**Published:** 2020-10-07

**Authors:** Fumi Mizuhashi, Takao Morita, Shuji Toya, Ritsuko Sato, Yuko Watarai, Kaoru Koide

**Affiliations:** 1Department of Removable Prosthodontics, the Nippon Dental University School of Life Dentistry at Niigata, Niigata 951-8580, Japan; watarai@ngt.ndu.ac.jp (Y.W.); koide@ngt.ndu.ac.jp (K.K.); 2Department of Biochemistry, the Nippon Dental University School of Life Dentistry at Niigata, Niigata 951-8580, Japan; moritat@ngt.ndu.ac.jp (T.M.); sator@ngt.ndu.ac.jp (R.S.); 3Dry Mouth Clinic, Oral and Maxillofacial Surgery, The Nippon Dental University Niigata Hospital, Niigata 951-8580, Japan; toya@ngt.ndu.ac.jp

**Keywords:** oral dryness, saliva, calcium blocker, protein ingredient

## Abstract

Oral dryness as a side effect of certain drugs is increasing. The aim of this study was to examine the change of the protein ingredient in saliva of oral dryness patients caused by calcium blocker. Six patients taking calcium blocker and six healthy elderly were enrolled. Unstimulated salivary flow rate, protein concentration, and flow rate of protein were measured and compared between the patients taking calcium blocker and healthy elderly. iTRAQ (Isobaric Tag for Relative and Absolute Quantitation) proteomic analysis was performed to extract the salivary protein changed in patient taking calcium blocker, and the intensities of Western blotting products were quantified (unpaired *t*-test). Unstimulated salivary flow rate was significantly lower on patients taking calcium blocker (*p* < 0.01). Protein concentration tended to be higher and the flow rate of protein tended to be lower on patients. As the result of iTRAQ proteomic analysis, calmodulin-like protein 3, glutathione S-transferase P, and keratin type I cytoskeletal 13 increased characteristically in patient taking calcium blocker, and the expression in calmodulin-like protein 3 was significantly larger (*p* < 0.01). The results of this study indicated that calmodulin-like protein 3 increased in patients taking calcium blocker and could be a salivary biomarker for oral dryness caused by calcium blocker.

## 1. Introduction

In this super-aging society, oral dryness patients have been increasing in Japan. Many elderly patients wearing dentures come to the hospital for treatment, and most of them have subjective oral dryness. One study reported that the proportion of oral dryness patients reached 0.9% to 64.8% [1]. Oral dryness can cause the caries, periodontal disease, pain by using denture, oral mucositis, taste disorder, and dysphagia [2,3,4,5,6], and can lead a decline in their quality of life.

The causes of oral dryness are salivary gland disease, Sjögren’s syndrome, diabetes mellitus, radiation therapy for head and neck cancer, and side effect of certain drugs. Most of elderly have some diseases, and taking medicines, and then the oral dryness caused by side effect of certain drugs are increasing. The kind of drugs that have a side effect of dry mouth extend to 700 to 800, and including antihypertensive drug, psychoactive drug, antihistaminic drug, diuretic, etc. [7,8,9,10]. There are many patients of hypertension in Japan, and 70% of the patients of hypertension taking calcium antagonist. One of the side effects of calcium antagonist is reported as dry mouth [11]; however, the mechanism that calcium antagonist cause dry mouth has not been cleared enough [12]. Additionally, it is well known that calcium antagonist causes gingival thickening, but there is not yet a unified view concerning the mechanism of pathogenesis [13,14]. We have hypothesized that calcium antagonist causes not only the calcium channel inhibition, but also some transcriptive activities because gingival thickening is led by proliferation of gingival constitution cell.

Saliva has not been used for the clinical examination because there were few reports concerning the clinical examination targeting saliva, and an easy appropriate method for saliva collecting was not established. Recently, it started to be reported that saliva which can be obtained without invasion has many bio-information equal in blood or urine [15]. It was clarified that saliva is non-conventional fluid associated with whole body hydration status, and the saliva flow rate was reduced and saliva total protein concentration was increased by dehydration [16]. Saliva composition could be collected from unstimulated saliva or stimulated saliva. Unstimulated saliva is less dependent of flow rate and pH, but sample volumes are lower than stimulated saliva. However, the stimulated saliva is much diluted [17]. Some sampling devices for saliva has been came onto the market, and the suitability of the collection method for the analyte of interest is important for the salivary biomarkers detection [17]. The collection procedures influence the saliva composition as one report indicated that sampling procedures of saliva influences the urate and the lactate concentration, and concentration in oral fluid decreased with the increase of the stimulation [18]. Thus, saliva could be a useful non-invasive substitute of blood if the sampling procedures were suitable.

The hypothesis of the present study was that the protein ingredient in saliva on oral dryness patients caused by calcium blocker differs from healthy elderly because of some transcriptive activities caused by calcium blocker. The purpose of this study was to test the following null hypothesis: the protein ingredient in saliva on oral dryness patients caused by calcium blockers was not different to that of healthy elderly.

## 2. Materials and Methods

### 2.1. Subjects

Oral dryness patients caused by calcium blocker (6 women, mean age: 70.3 ± 10.0 years, taking amlodipine besilate 2.5 mg over 5 years) who attended the Dry Mouth Clinic in The Nippon Dental University Niigata Hospital and healthy elderly without any systemic disease, any internal medicine, and any health problems (2 men, 4 women, mean age: 69.0 ± 5.3 years) who attended The Nippon Dental University Niigata Hospital for the periodic health examination were investigated. The study was conducted in accordance with the Declaration of Helsinki, and the protocol was approved by the Ethics Committee of The Nippon Dental University School of Life Dentistry at Niigata (ECNG-H-155). All subjects gave their informed consent for inclusion before participating in the study. Protocol of this study is shown in Figure 1.

### 2.2. Salivary Flow Rate and Protein Assay

Unstimulated whole saliva was collected by the spitting method using test tubes (Sarstedt AG & Co., Nümbrecht, Germany) by ejecting the whole saliva into the test tube over a 10-min period [19]. The amount of saliva was measured based on the scale on the test tube having a sequence of marks at 1-mL intervals, and 0.1-mL intervals are marked below 0.5 mL. The images were assessed using Photoshop (Adobe Systems, San Jose, CA, USA), and the amount of saliva was measured on the extended image. In the test tubes, 40 µL of a protease inhibitor cocktail (leupeptin 100 µM, trypsin inhibitor 75 µg/mL, 5 mM 4-amidinophenylmethane sulfonyl fluoride hydrochloride, 250 mM benzamidine, aprotinin 100 µg/mL) was already contained for the prevention of the degradation of the salivary proteins [20]. The saliva was collected between 9 to 11 a.m. The collected saliva was centrifuged at 14,000× *g* for 15 min at 4 °C to remove debris, and the supernatant was kept at −20 °C for further study (protein stability at −20 °C was already confirmed). The total protein concentration of the saliva was determined with a protein assay kit (Bio-Rad Laboratories, Hercules, CA, USA) with bovine serum albumin as a standard.

### 2.3. iTRAQ (Isobaric Tag for Relative and Absolute Quantitation) Proteomic Analysis

Saliva from one of the patients taking calcium blocker and one of the healthy elderly was analysed comparatively by iTRAQ proteomic analysis (Oncomics Co., Ltd., Nagoya, Japan) (Figure 2) for extracting the salivary protein changed in patient taking calcium blocker. All salivary protein in patients taking calcium blockers and the healthy elderly group was compared, and the salivary proteins significantly increased in patients taking the calcium blocker were extracted. 

### 2.4. Western Blot Analysis

The expression levels of salivary proteins that were increased on patient taking calcium blocker and β-Actin were examined using Western blot analyses. For the detection of each protein, the supernatant of the unstimulated whole saliva was used. The saliva protein samples (2 μg) were mixed with Laemmli sample buffer, and then subjected to 10% SDS-polyacrylamide gel electrophoresis (PAGE). After PAGE, the separated proteins were electrophoretically transferred from gel to PVDF membrane with iBlot Gel Transfer Stacks (Thermo Fisher Scientific, Waltham, MA, USA). The blots were separately probed with each specific primary antibody. The immune complexes were detected with horseradish peroxidase-conjugated secondary antibodies and SuperSignal West Dura substrate (Thermo Fisher Scientific) and visualized using Image Quant LAS500 (GE Healthcare, Fairfield, CT, USA). The intensities of each Western blotting product were quantified using NIH Image J (NIH, Bethesda, MD, USA).

### 2.5. Statistical Analysis

The difference in unstimulated salivary flow rate between the patients taking calcium blocker and healthy elderly was analyzed using unpaired *t*-test. The intensities of four Western blotting products were quantified; three products changed in patient taking calcium blocker and β-Actin as internal standard. The values of the three products changed in patient taking calcium blocker were divided by the value of β-Actin. The differences of the expression in salivary protein between the patients taking calcium blocker and healthy elderly were analyzed by unpaired *t*-test. Statistical analysis was performed using statistical analysis software (SPSS 17.0, SPSS JAPAN, Tokyo, Japan), and differences of α < 0.05 were considered significant.

## 3. Results

Table 1 shows the result of unstimulated salivary flow rate, protein concentration, and flow rate of protein on the patient taking calcium blocker and healthy elderly. Unstimulated salivary flow rate was significantly lower on patient taking calcium blocker than that on healthy elderly (*p* < 0.01). The protein concentration tended to be higher on patient taking calcium blocker than that on healthy elderly. The flow rate of protein tended to be lower on patient taking calcium blocker than that on healthy elderly.

As a result of iTRAQ proteomic analysis using one patient taking calcium blocker and one healthy elderly, three salivary proteins increased characteristically in patient taking calcium blocker were extracted; calmodulin-like protein 3, glutathione S-transferase P, and keratin type I cytoskeletal 13. On the six patients taking the calcium blocker and six healthy elderly patients, the intensities of the three Western blotting products were quantified and divided by the value of β-Actin. The expression in calmodulin-like protein 3 was statistically significantly different between the patients taking calcium blocker and healthy elderly (*p* < 0.01) (Figure 3). Figure 4 shows the detected protein bands of the calmodulin-like protein 3 on six patients taking calcium blocker and six healthy elderly.

## 4. Discussion

Oral dryness caused by side effect of certain drugs is increasing, and there are many patients taking calcium antagonists. However, the mechanism that calcium antagonist because dry mouth has not been cleared enough. We investigated the protein ingredient in saliva on oral dryness patients caused by calcium blocker and compared to the healthy elderly.

Unstimulated salivary flow rate in healthy elderly in this study was higher than that in patients taking calcium blocker. One report investigated the effect of anti-hypertensives on salivary flow, and resulted that unstimulated salivary flow rate in ACE inhibitor group was almost same value with healthy elderly, and that in beta blocker and diuretic agent groups was about five-sixths of healthy elderly people [21]. The results of this study showed that unstimulated salivary flow rate in patients taking calcium blocker was one eight of healthy elderly. These results suggested that the impact on oral dryness of calcium blocker would be significant in comparison with ACE inhibitor, beta blocker, and diuretic agent. The protein concentration tended to be higher on patients taking calcium blocker than that on healthy elderly people; on the other hand, the flow rate of protein tended to be lower on patients taking calcium blocker than that on healthy elderly people. The protein concentration on patients taking calcium blocker would be higher because of the low proportion of the watery discharge. This result could be explained by the previous study that clarified that a reduction in salivary flow rate and an increase in protein concentration were caused by whole-body dehydration [16]. Unstimulated salivary flow rate in patients taking calcium blocker was lower, therefore the flow rate of protein would be lower. These results suggested that the salivary flow rate in the patients taking calcium blocker became low with the low proportion of the watery discharge, and could cause oral dryness.

In this study, iTRAQ proteomic analysis was performed for extracting the salivary protein changed in patient taking calcium blocker. iTRAQ proteomic analysis is one of the exhaustive methods of the comparative quantitative analysis. Recently, some examinations started to be performed using iTRAQ proteomic analysis to identify salivary biomarkers for the diseases such as bisphosphonate-related osteonecrosis of the jaw [22], taste sensitivity [23], liver cancer [24], and oral squamous cell carcinomas [25]. As the result of iTRAQ proteomic analysis in this study, calmodulin-like protein 3, glutathione S-transferase P, and keratin type I cytoskeletal 13 were found to be increased characteristically in patients taking calcium blockers. The quantified protein bands of the calmodulin-like protein 3, glutathione S-transferase P, and keratin type I cytoskeletal 13 were divided by the value of β-Actin because β-Actin has a fixed manifestation and exists generally. Calmodulin-like protein compete with calmodulin, which plays a key role in the secretory function of saliva [26] and identified in Sjögren’s syndrome patients [27]. Glutathione S-transferases are enzyme family that involved in limiting oxidative damage to tissues [28], and reported more common in Sjögren’s syndrome [29]. Keratin is one of the proteins that have been investigated as disease parameters of Sjögren’s syndrome [30,31].

As the results of the statistical analysis, the expression in calmodulin-like protein 3 was statistically significantly different between the patients taking calcium blocker and healthy elderly. Calmodulin-like protein 3 is calcium-sensing receptor and many of them exist on epithelium [32]. Calmodulin-like protein is a protein which combines with calcium ion, and adjusts the cellular process by the interaction with specific protein that depends on calcium ion [33]. The reason why calmodulin-like protein 3 increased in patient taking calcium blocker could be considered that the calcium blocker acted to the calcium channel and inhibited the flow of calcium ion into the cell.

The results of this study suggested that there is a possibility that calmodulin-like protein 3 could be a salivary biomarker for oral dryness caused by calcium blocker. However, the main limitations of the study were that the number of the subjects was low, and the limited number of patients taking calcium blocker could influence the reliable conclusions because several additional factors can for surely modify the protein levels in saliva. Additionally, this study investigated only the patients of hypertension and only one kind of calcium antagonist. The sex of the patients was only women. These limited factors could bring some potential impacts on the protein content in saliva. In future studies, the biomarker for oral dryness should be examined with increasing subjects taking the calcium blocker.

## 5. Conclusions

The results of this study indicated that the protein ingredient in saliva on oral dryness patients caused by calcium blocker was different from that of healthy elderly. Calmodulin-like protein 3 increased in patient taking calcium blocker and could be a salivary biomarker for oral dryness caused by calcium blocker.

## Figures and Tables

**Figure 1 geriatrics-05-00070-f001:**
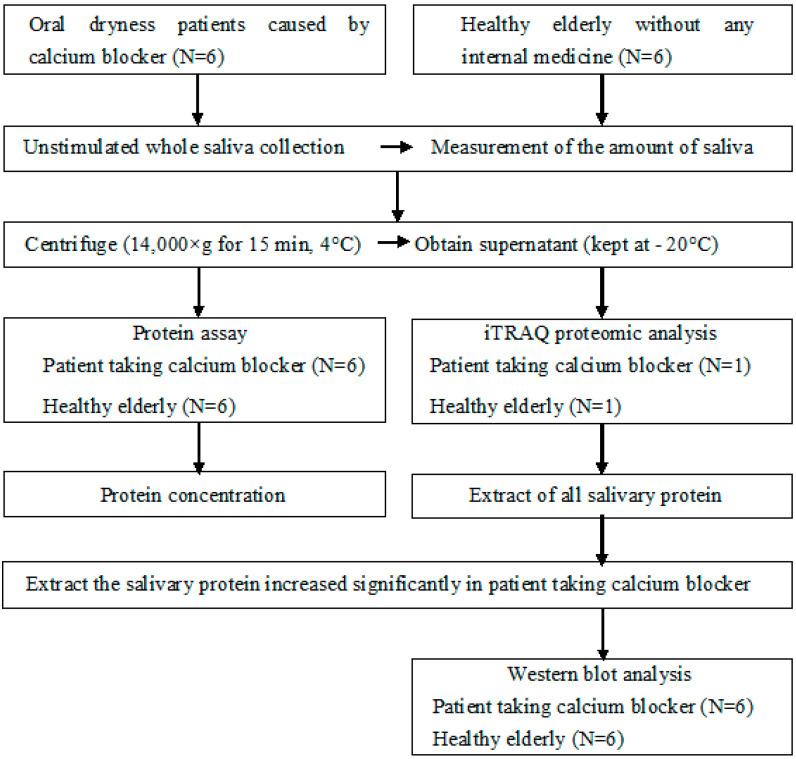
Protocol of this study. iTRAQ: Isobaric Tag for Relative and Absolute Quantitation.

**Figure 2 geriatrics-05-00070-f002:**
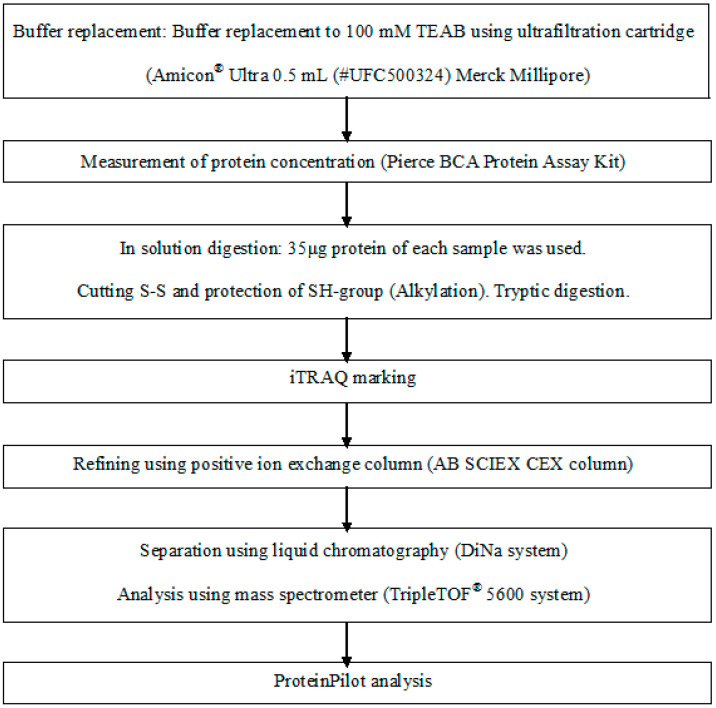
iTRAQ proteomic analysis. iTRAQ: Isobaric Tag for Relative and Absolute Quantitation.

**Figure 3 geriatrics-05-00070-f003:**
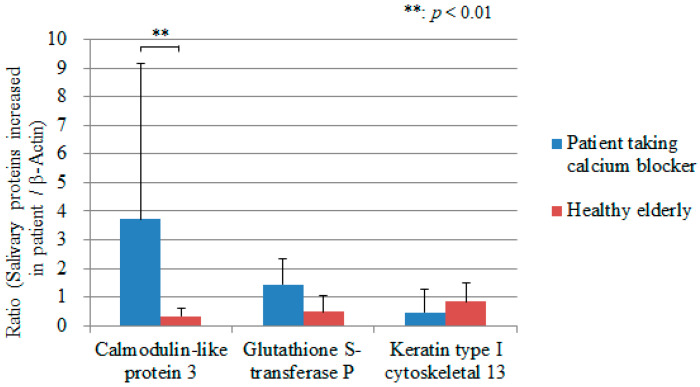
The expression of calmodulin-like protein 3, glutathione S-transferase P, and keratin type I cytoskeletal 13 divided by the value of β-Actin on the patient taking calcium blocker and healthy elderly. Measurements are expressed as mean value + SD. **: *p* < 0.01.

**Figure 4 geriatrics-05-00070-f004:**
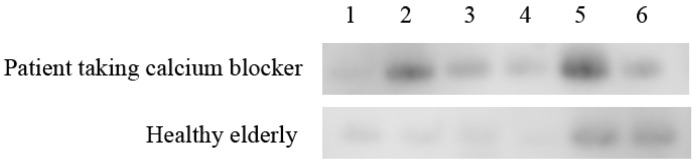
The detected protein bands of the calmodulin-like protein 3 on 6 patients taking calcium blockers and six healthy elderly.

**Table 1 geriatrics-05-00070-t001:** Unstimulated salivary flow rate, protein concentration, and flow rate of protein on patient taking calcium blocker and healthy elderly. Measurements are expressed as mean value ± SD.

	Patient Taking Calcium Blocker	Healthy Elderly
Unstimulated salivary flow rate (mL/10 min)	0.500 ± 0.191 *	4.108 ± 2.707 *
Protein concentration (μg/mL)	4729.63 ± 1642.24	1503.17 ± 533.96
Flow rate of protein (μg/mL)	244.67 ± 77.46	577.11 ± 323.01

* *p* < 0.01.

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
