# Peer review of "Protein Ingredient in Saliva on Oral Dryness Patients Caused by Calcium Blocker"

_geriatrics, 2020, doi:10.3390/geriatrics5040070_

Round 1
Reviewer 1 Report
The authors present the results of a small study examining the effect of calcium channel blockers on oral dryness. The introduction presents oral dryness as a growing problem in older adults, which they support with appropriate literature. The hypothesis is well spelled out. What would improve the manuscript is better description of the sample. Were the healthy older adults taking any medications—did they have any health problems? In addition, would the results be affected by the prevalence of females in the calcium channel blocker group versus 2 males, 4 females in the healthy sample?
Author Response
Response to Reviewer 1 Comments
Point 1: The authors present the results of a small study examining the effect of calcium channel blockers on oral dryness. The introduction presents oral dryness as a growing problem in older adults, which they support with appropriate literature. The hypothesis is well spelled out. What would improve the manuscript is better description of the sample. Were the healthy older adults taking any medications—did they have any health problems? In addition, would the results be affected by the prevalence of females in the calcium channel blocker group versus 2 males, 4 females in the healthy sample?
Response 1: Thank you very much for your comment. The healthy older adults were not taking any medications and did not have any health problems. Though most of the patients attending the Dry Mouth Clinic in our hospital were women, it remains to be seen if the results were affected by the prevalence of females in the calcium channel blocker. The number of the subjects in this study was small, therefore, in the future study; we should examine with increasing the subjects taking calcium blocker. The Materials and Methods was revised like “(line number: 73 - 74) healthy elderly without any systemic disease, any internal medicine, and any health problems (2 men, 4 women, mean age: 69.0±5.3 years)” The Discussion was revised like “(line number: 196 - 202) However, main limitations of the study was that the number of the subjects was small, and the limited number of patients taking calcium blocker could influence the reliable conclusions because several additional factors can for sure modify the protein levels in saliva. Additionally, this study investigated only the patients of hypertension and only one kind of calcium antagonist. The sex of the patients was only women. These limited factors could bring some potential impacts on the protein content in saliva.”
Reviewer 2 Report
Dear Authors, the article is well written and organised. However, the main problem is the number of patients enrolled in this pilot study, i.e. two groups of 6 patients. Unfortunately, the limited number of patients is not adequate for a robust and reliable conclusions. Several additional factors can for sure modify the protein levels in saliva and thus I strongly recommend to perform a power analysis with the aim to calculate the right number of patients to be enrolled.
Please find below some comments on the text:
- Introduction section, I suggest to extend the concept of saliva analysis, focusing the attention on the peculiar advantages of such fluid over conventional specimens. Moreover, the impact of the collection procedures on the saliva composition should be also discussed in the introduction. The following articles can be useful for the authors:
10.1016/j.microc.2017.02.032
doi.org/10.1016/j.trac.2019.115781
10.1016/j.archoralbio.2003.08.001 - L60, how was the variation of the calcium blocker dose? Did you find a potential relationship between the protein level and the dosage?
- L68, please explain better the type of the collection procedure used. It is not clear if the authors used the swallowing method or split method. How was precise and accurate the scale on the tube test? This is extremely important since the amount of saliva collected allows to estimate the salivary flow rate.
- L75, did you evaluate the protein stability at -20°C?
- L77, please include the analytical figure of merits of the protein assay kit.
- L79, please explain better the meaning of "... from a typical individual..." So, did you pooled all the saliva samples? Or did you analysed only a subset of the saliva samples collected from the patients? Again, please include all the analytical figure of merits.
- Table 2, please modify the significant digits of the protein concentration and flow rate of protein. The numbers should be consistent with the methods variability.
- L131.137, please explain if these analysed were carried out on all the saliva samples or on just few samples.
- Figure 1, please explain the ratio on the y-axis and introduce the error bar for each column.
- Discussion, please discussion the main limitations of the study and the potential impact of additional variables (diseases, other drugs, sex and etc) on the protein content in saliva.
Author Response
Response to Reviewer 2 Comments
Dear Authors: The article is well written and organised. However, the main problem is the number of patients enrolled in this pilot study, i.e. two groups of 6 patients. Unfortunately, the limited number of patients is not adequate for a robust and reliable conclusions. Several additional factors can for sure modify the protein levels in saliva and thus I strongly recommend to perform a power analysis with the aim to calculate the right number of patients to be enrolled.
Response to Dear Authors: Thank you very much for your comment. As you mentioned, the number of patients enrolled in this study was small. The number of the patients taking only calcium blocker was so small, and many patients are taking plural medicines. Additionally, there are not so many healthy elderly over 65 years old without any systemic disease and any internal medicine. Most of the healthy elderly are taking medicine. Therefore, in this study, the number of patients was still small. We are trying to increase the number of the patients taking only calcium blocker and healthy elderly over 65 years old without any systemic disease and any internal medicine. In the future study, we should investigate the protein ingredient in saliva on oral dryness patients caused by calcium blocker with increased number. We mentioned the limitation of this study in Discussion like “(line number: 196 - 203) However, main limitations of the study was that the number of the subjects was small, and the limited number of patients taking calcium blocker could influence the reliable conclusions because several additional factors can for sure modify the protein levels in saliva. Additionally, this study investigated only the patients of hypertension and only one kind of calcium antagonist. The sex of the patients is only women. These limited factors could bring some potential impacts on the protein content in saliva. In the future study, the biomarker for oral dryness should be examined with increasing the subjects taking calcium blocker.”
Point 1: Introduction section, I suggest to extend the concept of saliva analysis, focusing the attention on the peculiar advantages of such fluid over conventional specimens. Moreover, the impact of the collection procedures on the saliva composition should be also discussed in the introduction. The following articles can be useful for the authors:
10.1016/j.microc.2017.02.032
doi.org/10.1016/j.trac.2019.115781
10.1016/j.archoralbio.2003.08.001
Response 1: Thank you very much for your suggestion, and thank you so much for showing the useful articles. We referred to these articles and extended the concept of saliva analysis by mentioning that saliva is non-invasive markers of whole body hydration status. Additionally, we discussed the impact of the collection procedures on the saliva composition. The sentence was inserted in the Introduction like “(line number: 52 - 63) It was clarified that saliva is non-invasive markers of whole body hydration status, and the saliva flow rate was reduced and saliva total protein concentration was increased by dehydration [16]. Saliva composition could be collected from unstimulated saliva or stimulated saliva. Unstimulated saliva is less dependent of flow rate and pH, but sample volumes are lower than stimulated saliva. However, the stimulated saliva is much diluted [17]. Some sampling devices for saliva has been came onto the market, and the suitability of the collection method for the analyte of interest is important for the salivary biomarkers detection [17]. The collection procedures influences the saliva composition as one report indicated that sampling procedures of saliva influences the urate and the lactate concentration, and concentration in oral fluid decreased with the increase of the stimulation [18]. Thus, saliva could be a useful non-invasive substitute of blood if the sampling procedures were suitable.” The references [16-18] were also added (line number: 249 - 257).
16. Walsh, N.P.; Montague, J.C.; Callow, N.; Rowlands, A.V. Saliva flow rate, total protein concentration and osmolality as potential markers of whole body hydration status during progressive acute dehydration in humans. Arch. Oral Biol. 2004, 49, 149-154.
17. Bellagambi, F.G.; Lomonaco, T.; Salvo, P.; Vivaldi, F.; Hangouët, M.; Ghimenti, S.; Biagini, D.; Francesco, F.D.; Fuoco, R.; Errachid, A. Saliva sampling: Methods and devices. An overview. Trends Analyt. Chem. 2020, 124, 115781.
18. Lomonaco, T.; Ghimenti, S.; Biagini, D.; Bramanti, E.; Onor, M.; Bellagambi, F.G.; Fuoco, R.; Francesco, F.D. The effect of sampling procedures on the urate and lactate concentration in oral fluid. Microchem. J. 2018, 136, 255-262.
Point 2: L60, how was the variation of the calcium blocker dose? Did you find a potential relationship between the protein level and the dosage?
Response 2: Thank you very much for your suggestion. All of the 6 patients were taking amlodipine besilate 2.5mg over 5 years. We have not found a potential relationship between the protein level and the dosage. As you commented, relationship between the protein level and the dosage should be investigated. In the future study, we are going to investigate the relationship between the protein level and the dosage. The Materials and Methods was revised like “(line number: 71 - 72) Oral dryness patients caused by calcium blocker (6 women, mean age: 70.3±10.0 years, taking amlodipine besilate 2.5mg over 5 years)…”
Point 3: L68, please explain better the type of the collection procedure used. It is not clear if the authors used the swallowing method or split method. How was precise and accurate the scale on the tube test? This is extremely important since the amount of saliva collected allows to estimate the salivary flow rate.
Response 3: Thank you very much for your suggestion. We used the spitting method for the collection of whole saliva. The test tubes (Sarstedt AG & Co., Nümbrecht, Germany) having a sequence of marks at 1ml intervals, and 0.1ml intervals are marked below 0.5ml. We decided the amount of saliva on the image magnified. The Materials and Methods was revised like “(line number: 81 - 86) Unstimulated whole saliva was collected by the spitting method using test tubes (Sarstedt AG & Co., Nümbrecht, Germany) by ejecting the whole saliva into the test tube over a 10-min period [19]. The amount of saliva was measured based on the scale on the test tube having a sequence of marks at 1ml intervals, and 0.1ml intervals are marked below 0.5ml. The images were assessed using Photoshop (Adobe Systems, San Jose, CA, USA), and the amount of saliva was measured on the extended image.”
Point 4: L75, did you evaluate the protein stability at -20°C?
Response 4: Thank you very much for your comment. Yes, we evaluated the protein stability at -20°C, and confirmed that the protein was stable at -20°C. The Materials and Methods was revised like “(line number: 90 - 91) the supernatant was kept at −20°C for further study (protein stability at -20°C was already confirmed)”
Point 5: L77, please include the analytical figure of merits of the protein assay kit.
Response 5: Thank you very much for your suggestion. We added the Table 1 as the analytical figure of merits to explain the method of this study. The Table 1 was added to the Materials and Methods (line number: 79).
Point 6: L79, please explain better the meaning of "... from a typical individual..." So, did you pooled all the saliva samples? Or did you analysed only a subset of the saliva samples collected from the patients? Again, please include all the analytical figure of merits.
Response 6: Thank you very much for your comment. We extracted at random one saliva from the 6 patients taking calcium blocker and one saliva from the 6 healthy elderly for iTRAQ proteomic analysis. The expression of "... from a typical individual..." was inadequate, and revised like “(line number: 95 - 96) Saliva from one of the patients taking calcium blocker and one of the healthy elderly was analysed comparatively by iTRAQ proteomic analysis” We added the Table 1 as the analytical figure of merits to explain the method of this study. The Table 1 was added to the Materials and Methods (line number: 79).
Point 7: Table 2, please modify the significant digits of the protein concentration and flow rate of protein. The numbers should be consistent with the methods variability.
Response 7: Thank you very much for your comment. As you mentioned, the significant digits of the protein concentration and flow rate of protein should be consistent with the methods variability. We modified the significant digits of the protein concentration and flow rate of protein on Table 3 (Table 2 in initial submission). Table 3 was replaced (line number: 133). Abstract was revised like “(line number: 21 - 23) Unstimulated salivary flow rate was significantly lower on patients taking calcium blocker (p < 0.01). Protein concentration tended to be higher and the flow rate of protein tended to be lower on patients.” The Materials and Methods was revised like “(line number: 118 - 119) The difference in unstimulated salivary flow rate between the patients taking calcium blocker and healthy elderly was analysed using unpaired t-test.” and the Results was revised like “(line number: 129 - 132) The protein concentration tended to be higher on patient taking calcium blocker than that on healthy elderly. The flow rate of protein tended to be lower on patient taking calcium blocker than that on healthy elderly.”
Point 8: L131.137, please explain if these analysed were carried out on all the saliva samples or on just few samples.
Response 8: Thank you very much for your suggestion. iTRAQ proteomic analysis was carried out on one sample of taking calcium blocker and one sample of healthy elderly to extract possible proteins changed in patient taking calcium blocker. Figure 2 shows the detected protein bands carried out on all the saliva samples. We added the Table 1 as the analytical figure of merits to explain the method of this study with the number of saliva sample included. The Results was revised like “(line number: 135 - 136) As the result of iTRAQ proteomic analysis using one patient taking calcium blocker and one healthy elderly,…” and “(line number: 141 - 143) Figure 2 shows the detected protein bands of the calmodulin-like protein 3 on six patients taking calcium blocker and six healthy elderly.”
Point 9: Figure 1, please explain the ratio on the y-axis and introduce the error bar for each column.
Response 9: Thank you very much for your suggestion. We added the explanation of the ratio on the y-axis and introduce the error bar for each column. The Figure 1 was replaced (line number: 144).
Point 10: Discussion, please discussion the main limitations of the study and the potential impact of additional variables (diseases, other drugs, sex and etc) on the protein content in saliva.
Response 10: Thank you very much for your suggestion. The main limitations of the study were that the number of patients enrolled in this study was small. The potential impact of additional variables was that this study investigated only one kind of disease and one kind of drug. Additionally, the sex of the patients was only women. The Discussion was revised like “(line number: 196 - 202) However, main limitations of the study was that the number of the subjects was small, and the limited number of patients taking calcium blocker could influence the reliable conclusions because several additional factors can for sure modify the protein levels in saliva. Additionally, this study investigated only the patients of hypertension and only one kind of calcium antagonist. The sex of the patients was only women. These limited factors could bring some potential impacts on the protein content in saliva.”
Round 2
Reviewer 2 Report
Dear Authors, some minor modifications are needed before to accept the paper.
L53, please check the sentence. Saliva is not a marker but a non-conventional fluid. I suggest to rewrite the sentence in a proper manner.
Table 3, please change the digit numbers according to the overall method variability.
Figure 1, please explain in the caption the error bars.
